# Fluoroscopic 3D Image Generation from Patient-Specific PCA Motion Models Derived from 4D-CBCT Patient Datasets: A Feasibility Study

**DOI:** 10.3390/jimaging8020017

**Published:** 2022-01-18

**Authors:** Salam Dhou, Mohanad Alkhodari, Dan Ionascu, Christopher Williams, John H. Lewis

**Affiliations:** 1Department of Computer Science and Engineering, College of Engineering, American University of Sharjah, Sharjah 26666, United Arab Emirates; 2Healthcare Engineering Innovation Center (HEIC), Department of Biomedical Engineering, Khalifa University, Abu Dhabi 127788, United Arab Emirates; mohanad.alkhodari@ku.ac.ae; 3Department of Radiation Oncology, College of Medicine, University of Cincinnati, Cincinnati, OH 45267, USA; ionasctn@ucmail.uc.edu; 4Department of Radiation Oncology, Brigham and Women’s Hospital and Harvard Medical School, Boston, MA 02115, USA; cwilliams@bwh.harvard.edu; 5Cedars-Sinai Medical Center, Los Angeles, CA 90048, USA; john.h.lewis@cshs.org

**Keywords:** principal component analysis (PCA), motion model, respiratory-correlated four-dimensional cone-beam CT (4D-CBCT), lung cancer, stereotactic body radiotherapy (SBRT), image-guided radiation therapy (IGRT)

## Abstract

A method for generating fluoroscopic (time-varying) volumetric images using patient-specific motion models derived from four-dimensional cone-beam CT (4D-CBCT) images was developed. 4D-CBCT images acquired immediately prior to treatment have the potential to accurately represent patient anatomy and respiration during treatment. Fluoroscopic 3D image estimation is performed in two steps: (1) deriving motion models and (2) optimization. To derive motion models, every phase in a 4D-CBCT set is registered to a reference phase chosen from the same set using deformable image registration (DIR). Principal components analysis (PCA) is used to reduce the dimensionality of the displacement vector fields (DVFs) resulting from DIR into a few vectors representing organ motion found in the DVFs. The PCA motion models are optimized iteratively by comparing a cone-beam CT (CBCT) projection to a simulated projection computed from both the motion model and a reference 4D-CBCT phase, resulting in a sequence of fluoroscopic 3D images. Patient datasets were used to evaluate the method by estimating the tumor location in the generated images compared to manually defined ground truth positions. Experimental results showed that the average tumor mean absolute error (MAE) along the superior–inferior (SI) direction and the 95th percentile in two patient datasets were 2.29 and 5.79 mm for patient 1, and 1.89 and 4.82 mm for patient 2. This study demonstrated the feasibility of deriving 4D-CBCT-based PCA motion models that have the potential to account for the 3D non-rigid patient motion and localize tumors and other patient anatomical structures on the day of treatment.

## 1. Introduction

Respiratory-induced organ motion is a major source of uncertainty in stereotactic body radiotherapy (SBRT) of thoracic and upper abdominal cancers [1]. Respiratory motion can result in motion artifacts during image acquisition and limitations in both radiotherapy planning and delivery. Respiratory-correlated, or four-dimensional (4D) computed tomography (4DCT), as an image-guided radiation therapy (IGRT) tool, provides a solution to obtain high quality CT images in the presence of respiratory motion [2]. Thus, 4DCT became a standard method in radiotherapy treatment planning to account for organ motion, reduce motion artifacts, and reduce associated uncertainties.

Image-based motion modeling of patient anatomy during radiotherapy can be useful in accurately localizing tumors and other anatomical structures in the body [3,4,5,6,7]. There are many approaches proposed for image-based motion modeling. Principal component analysis (PCA)-based motion modeling has proven its efficacy in representing the spatio-temporal relationship of the entire lung motion [8]. Because of their compactness and performance, PCA motion models are being used along with projection images captured at the day of treatment for generating time-varying volumetric images, often called fluoroscopic because they are produced in a continuous fashion similar to the images produced using the well-known fluoroscopy procedure [9,10,11,12,13,14,15,16]. PCA motion models are derived by applying PCA on the displacement vector fields (DVFs) that result from applying deformable image registration (DIR) between the 4DCT phases and a reference phase chosen from the same set. PCA distills the large dataset of DVFs into a few eigenvectors and coefficients representing lung motion [8,12,14,16,17]. Because 4DCT images are acquired at the time of treatment planning, which happens days or weeks before the treatment delivery day, PCA motion models derived from them may not accurately represent patient anatomy or motion patterns at the day of treatment delivery [14]. Consequently, they may not account for tumor baseline shifts that are observed frequently in the clinic [18].

Respiratory-correlated, or four-dimensional (4D) cone-beam CT (4D-CBCT), has been introduced and used in radiotherapy for many clinical tasks such as image guidance and target verification just prior to treatment delivery [19]. 4D-CBCTs are reconstructed by first assigning the raw CBCT projections into several bins depending on the respiratory phases they exhibit; then, 3D images are reconstructed from each bin. Several methods have been used to estimate respiratory motion corresponding to the raw CBCT projections. These methods include using external equipment, such as external markers or abdominal belts, internal implanted radiopaque fiducial markers, or marker-free pure image-based approaches [20,21,22,23,24,25,26,27]. On-board 4D-CBCT images are produced at the day of treatment delivery while the patient is in treatment position. Thus, motion models derived from 4D-CBCT images have the potential to account for the inter-fraction anatomical motion variations that can occur between the planning and treatment delivery phases, which may not be handled using 4DCT-based motion models.

Previous research has been conducted to derive PCA motion models from 4D-CBCT images [9,10,28,29]. In [9,10], PCA motion models were derived from 4D-CBCT datasets of simulated patients using the digital 4D Extended Cardiac-Torso (XCAT) phantom and an anthropomorphic physical phantom. In these studies, a total of eight 4D-CBCT datasets were simulated using XCAT software [30,31,32]. These datasets featured different tumor locations and breathing signals measured from lung cancer patients. Moreover, two 4D-CBCT datasets were simulated by taking CBCT images of an anthropomorphic physical phantom, which was a modified version of the Alderson Lung/Chest Phantom (Radiology Support Devices, Inc., Long Beach, CA, USA). In the phantom’s rib cage, foam slices were inserted to simulate lung tissue and to hold a tumor model. During the image acquisition, the foam slices were pushed in and out using a programmable translation stage in the superior–inferior (SI) direction to simulate diaphragm motion and breathing. PCA motion models were derived from all these phantom datasets and used to generate fluoroscopic 3D images. These studies showed the feasibility and reliability of estimating anatomical motion using 4D-CBCT-based motion models compared to 4DCT-based motion models. However, the experiments were only applied to phantom datasets, and hence the efficacy of this approach on clinical patient datasets has not been verified. In the other studies [28,29], PCA motion models were derived from datasets of patients’ 4D-CBCT images taken at different treatment days to quantify the inter-fraction variations of these motion models. However, these 4D-CBCT-based PCA motion models were not used in further clinical tasks such as generating fluoroscopic 3D images or localizing tumors and/or other anatomical structures of the patients at the time of treatment delivery. 

In this study, we proposed to: (1) derive PCA motion models from patient 4D-CBCT images captured immediately before treatment delivery; and (2) use these 4D-CBCT-based motion models to estimate fluoroscopic 3D images based on CBCT projections captured immediately before treatment delivery. The proposed work is an extension to the previous work [9,10] where the methods were tested on digital phantom datasets and anthropomorphic physical phantom datasets. In this study, the methods were applied on patient datasets to demonstrate the feasibility of considering this approach in clinical settings. The rest of this paper is organized as follows. Section 2 discusses the materials and methods used in this work. Section 3 presents the experimental results. The results are discussed in Section 4. Section 5 concludes the paper.

## 2. Materials and Methods

### 2.1. Datasets

CBCT projections for two patients were acquired using the Elekta Synergy system (Elekta Oncology Systems Ltd., Crawley, UK) and used retrospectively in this study. This retrospective research protocol qualified for exempt approval from the Institutional Review Board (IRB) of the American University of Sharjah, United Arab Emirates, on 23 August 2021 (IRB 18-425). The projections were acquired over 200 degree rotations at 5.5 fps in 4 min. The total number of projections is 1320 in the first patient dataset and 1356 in the second patient dataset. The dimensions of the projections are 512 × 512 pixels in both datasets. 4D-CBCT images were reconstructed from each projection dataset. To do so, the projections were sorted into six phase bins according to their corresponding respiratory status estimated using the “Amsterdam Shroud” method [33,34]. The Feldkamp, Davis, and Kress (FDK) reconstruction algorithm [35] implemented in Reconstruction ToolKit (RTK) [36] was used to reconstruct 3D images from each projection bin, which resulted in 4D-CBCT images of six phases. The dimensions of each of the reconstructed images are 176 × 228 × 256 voxels, with 1.1 as the voxel size.

The ground truth tumor location for the patient datasets was found by manually identifying the diaphragm location in each projection. A simple graphical user interface was programmed in MATLAB (The MathWorks, Inc., Natick, MA, USA) and used for that purpose. To estimate the coordinates of the tumor in each projection, the diaphragm apex coordinates were identified in each projection and used in a linear regression model to estimate the tumor coordinates. To compare the ground truth 2D coordinates with the tumor 3D coordinates in the estimated fluoroscopic 3D images, the tumor coordinates in the estimated fluoroscopic 3D images were projected onto a 2D flat panel detector. The distance between the 2D projected and ground truth coordinates was calculated in the plane of the detector, and then scaled down to an approximate error inside the patient (at isocenter). A similar procedure was followed in previous publications [10,16,37].

Figure 1 shows axial, coronal, and sagittal slices of peak-exhale 4D-CBCT from each patient. The peak-exhale phase was selected as the reference phase to which all other 4D-CBCT phases are deformed in the DIR module.

### 2.2. Fluoroscopic 3D Image Estimation 

Fluoroscopic 3D image estimation using PCA motion models is a well-known approach that has been used in several previous studies [9,10,11,12,13,14,15,16]. In this work, the same approach is used but the input to this algorithm is the 4D-CBCT images for real patients. Fluoroscopic 3D image estimation algorithm is accomplished in two steps as follows:

#### 2.2.1. 4D-CBCT-Based Motion Model Estimation

To estimate the motion models, DIR is applied to each 4D-CBCT phase with respect to a reference phase chosen from the same set. In this work, the peak-exhale phase was chosen as the reference phase. Demon’s DIR algorithm implemented on a graphics processing unit (GPU) was used in this study [38]. This algorithm was considered in this work because it is a non-rigid registration algorithm that has been used extensively in the literature to register 3D medical images. Moreover, this algorithm has been used in previous fluoroscopic 3D image generation studies similar to this work, which proved that the error caused by this algorithm is negligible [16].

Applying DIR on pairs of phases results in a set of DVFs describing the voxel-wise displacements between each pair of phases. As the resulting DVFs represent a huge dataset, a dimensionality reduction approach is used to transform this dataset from the original high-dimensional space into a low-dimensional one while retaining the properties of the original data. PCA was employed as a linear dimensionality reduction method. PCA is applied on the DVFs which results in a set of eigenvectors and eigenvalues representing the motion of the patient [8]. The set of DVFs can be represented as a weighted sum of these eigenvectors and eigenvalues as follows:(1)$$D=\bar{D}+\sum_{i=1}^{N}v_{i}\;u_{i}\left(t\right),$$
where D is the DVF dataset, $\bar{D}$ is the mean DVF, $u_{i}\left(t\right)\;$ represents the PCA eigenvalues defined in time, $v_{i}$ represents the eigenvectors defined in space, and *N* is the number of eigenmodes considered. The eigenvectors can be sorted according to their corresponding eigenvalues such that eigenvectors corresponding to the largest eigenvalues represent a large fraction of the variance of the original data. Previous studies have shown that the first few (2–3) eigenvectors, corresponding to the largest eigenvalues, are sufficient to represent the motion patterns existing in the original dataset [10,12,29]. In this work, the first three eigenvectors were considered as the motion model.

#### 2.2.2. Optimization

An optimization approach is used to estimate the fluoroscopic 3D images. This approach involves three inputs: (1) the motion model (a set of 3 eigenvectors with corresponding eigenvalues); (2) the 4D-CBCT reference phase; and (3) the CBCT projections captured immediately before treatment, while the patient is in treatment position. The working principle of this optimization approach is to iteratively update the motion model by minimizing a cost function representing the squared L_2_-norm of the difference between a CBCT projection captured at treatment time and a 2D projection computed using both the motion model and the 4D-CBCT reference phase. The cost function is represented by:(2)$$\underset{u}{\min}J\left(u\right)=\|P\cdot f\left(D\left(u\right),\;f_{0}\right)-\lambda \cdot x\|$$
where $f_{0}$ is the 4D-CBCT reference phase, D(u) represents the parameterized DVFs, f is the estimated fluoroscopic 3D image, P is the projection matrix used to compute the projection from the fluoroscopic 3D image f, x is the CBCT projection captured at treatment delivery day, and λ is the relative pixel intensity between the 2D computed projection and the CBCT projection x. The cost function is minimized using a version of gradient descent, as explained in the appendix in [16]. Figure 2 presents the flowchart of the fluoroscopic 3D image estimation algorithm. As can be seen from the figure, the difference between this study and the previous fluoroscopic 3D image estimation studies is that the input to this study is the 4D-CBCT images acquired for patients at treatment delivery time. This approach will result in deriving 4D-CBCT-based PCA motion models that can be used to estimate fluoroscopic images for patients on the treatment delivery day. 

In this approach, a linear relationship between the intensities of the CBCT projections and the computed projections using the motion model is assumed. However, some factors may disturb this assumption, such as noise and the poor quality of the 4D-CBCT images being used to compute the 2D projections. The limited number of projections available for reconstruction of each 4D-CBCT bin can cause artifacts that can appear as streaks in the resulting 4D-CBCT images. These artifacts may cause non-anatomical differences between the real CBCT projections and the corresponding 2D projections computed using the 4D-CBCT-based motion model. Thus, considering a region of interest (ROI) surrounding the tumor and other moving anatomical structures in the images, such as the diaphragm apex, has the potential to reduce the effect of these differences in the optimization procedure. In this work, a ROI was chosen from both the CBCT projection and the corresponding 2D computed projection to reduce the effect of the noise and artifacts existing in the whole images and enhance the accuracy of the optimization. The ROI was chosen to surround the tumor and the diaphragm apex, which are the most visible structures in the image exhibiting breathing motion.

### 2.3. Evaluation

The method was evaluated by finding the tumor localization error, which is calculated as the mean absolute error (MAE) of the tumor centroid location in the estimated fluoroscopic 3D images. This error value is measured by taking the mean absolute difference between the tumor centroid location in the estimated fluoroscopic 3D images and the ground truth locations. The process of estimating ground truth tumor coordinates is described in Section 2.1. The 3D tumor coordinates in the estimated fluoroscopic 3D images are projected onto a 2D flat panel detector to be able to compare them with the 2D ground truth tumor coordinates. The error is measured along the superior–inferior (SI) direction in patient coordinates.

## 3. Results

In this section, the estimated PCA motion models and fluoroscopic 3D images are evaluated. Firstly, to evaluate the PCA motion models, an explained variance study was carried out. Explained variance is a statistical analysis study that is used to measure the proportion of the variation of a given dataset that is accounted for by a mathematical model. In this work, the analysis was carried out to explore the variance explained by each PCA eigenvector and to determine the number of PCA eigenvectors that can be considered in the motion model without losing important information. Figure 3a shows the eigenvalue spectrum for the PCA motion models for patient #1 and patient #2. It can be observed that the eigenvalues decrease with higher eigenmodes and drop drastically after the third eigenmode in both patients. Figure 3b shows the percentage of the variance explained by each eigenmode in each patient. It shows both the individual and cumulative explained variances. As can be seen from the figure, the first three eigenvectors can explain most of the variance (97.1% in patient #1) and (97.4% in patient #2).

The PCA motion models derived from 4D-CBCT images were used to estimate the fluoroscopic 3D images. Firstly, a correlation analysis was conducted to measure the correlation between the intensities of a sample CBCT projection and the corresponding 2D projection computed using the estimated motion model and the 4D-CBCT reference phase. This study is important to prove the linear relationship between the intensities of the two projections as assumed in the cost function described by Equation (2). Figure 4 shows a CBCT projection and the corresponding 2D projection computed using the estimated motion model and the 4D-CBCT reference phase. A scatter plot showing the linear correlation between the intensities of the two images is shown. As can be seen from the figure, a linear correlation was found with a correlation coefficient of 96%.

The estimated fluoroscopic 3D images for each of the datasets were evaluated. Figure 5 shows axial, coronal, and sagittal slices of a sample estimated fluoroscopic 3D image from patient #2. Figure 6 shows coronal slices of two estimated fluoroscopic 3D images from patient #2 at different breathing phases. As can be noticed from Figure 6, the estimated fluoroscopic images were able to capture the anatomical motion represented in the CBCT projections used in the optimization module.

To evaluate the accuracy of the estimated fluoroscopic images, the SI tumor position in the estimated images was measured and compared to its ground truth location. Figure 7 shows the SI tumor position in all the estimated fluoroscopic 3D images compared to the ground truth tumor positions in millimeters for patient #1 (a) and patient #2 (b). The tumor MAE along the SI direction was 2.29 mm with a 95th percentile of 5.79 mm for patient #1, and 1.89 mm with a 95th percentile of 4.82 mm for patient #2.

## 4. Discussion

In this work, the feasibility of building patient-specific motion models from 4D-CBCT images and using them, along with a set of CBCT projections captured at the time of treatment delivery, to generate fluoroscopic 3D images was studied. The 4D-CBCT-based motion models have the potential to overcome an important shortcoming of the 4DCT-based motion models in that they can reflect the patient anatomy and motion at the time of treatment delivery. These fluoroscopic 3D images can be used in several clinical applications such as delivered dose verification [39,40].

The methodology used in this work involved two main steps: deriving the PCA motion model and the optimization approach to estimate the fluoroscopic 3D images. The method was evaluated on two patient datasets. The resulting PCA 4D-CBCT-based motion models that were used in this work were analyzed in Section 3. As mentioned in Figure 3, the first few eigenmodes of these PCA motion models explained most of the variance in the DVF dataset. Based on this, the remainder of the eigenmodes were dropped safely as they do not hold significant information. These results support the findings of other studies that showed that a small number of eigenmodes (2–3) are sufficient to represent the organ motion represented by the DVFs [10,12,16,29]. The iterative optimization approach was shown to converge after several iterations, which resulted in producing optimized fluoroscopic 3D images representing the anatomical motion of the patient. The algorithm was implemented to run efficiently on a GPU (NVIDIA GeForce GTX 1070, 8 GB VRAM). The DIR algorithm takes an average of 17.25 s to register a 4D-CBCT phase to the reference phase. The optimization step needs an average of 1.25 s to estimate a fluoroscopic 3D image, including the time required to estimate the tumor location. 

Comparing this error to other studies using 4D-CBCT-based motion models derived from phantom datasets [10], it can be noticed that the error in this study is slightly higher. Given the complexities of the breathing patterns of the real patients and the poor quality of the 4D-CBCT images, having a higher tumor error is expected. DIR accuracy is a key determinant of motion model accuracy. DIR yields the DVFs upon which the motion model is based. The DIR algorithm used in this study is the Demon’s algorithm [38]. In a previous fluoroscopic 3D image generation study using 4DCT images, the authors investigated the effect of DIR performance on the overall method accuracy [16]. It was observed that the error caused by DIR is negligible. In that study, the error was mainly attributed to optimization step of the method—specifically, the mapping between the CBCT projection and the computed one. 

One of the major challenges facing the construction of motion models from 4D-CBCT images is the poor quality of the 4D-CBCT images used as input. The limited number of projections available for reconstruction of each 4D-CBCT bin are a key reason for the relatively poor quality of 4D-CBCT images. The effect of the quality of the 4D-CBCT images in fluoroscopic 3D image estimation was investigated in [10]. The authors conducted an experiment using two sets of 4D-CBCT images simulated using a digital XCAT phantom. The 4D-CBCT images in the first set were reconstructed from a well-sampled set of projections, whereas the images in the second set were reconstructed from a severely under-sampled set of projections. The study showed that the tumor MAE along the SI direction increased by 214% (from 1.28 to 4.02 mm) with a 95th percentile increasing by 250% (from 2.0 to 7.00 mm) when using the 4D-CBCT images that were reconstructed from an under-sampled set of projections as the input to this method. The normalized root mean square error (NRMSE) calculated using the voxel-wise intensity difference between the resulting estimated images and the ground truth images also increased by 150% (from 0.10 to 0.25 mm) when using the 4D-CBCT images that were reconstructed from an under-sampled set of projections. The under-sampling issue in 4D-CBCT has been studied extensively in the literature. Several solutions to improve the quality of the 4D-CBCT images have been suggested, such as compressed sensing [41,42,43,44,45], motion compensated reconstruction [46,47,48,49,50,51,52,53,54], and interpolation of “in-between” projections to increase the number of projections in each respiratory phase bin [55,56,57,58]. Recently, deep learning approaches have also been proposed [59,60,61]. Motion modeling and fluoroscopic image estimation from enhanced 4D-CBCT images is worth investigating in future research.

## 5. Conclusions

This study investigated the feasibility of deriving motion models from patient 4D-CBCT images and using them to generate fluoroscopic 3D images of the patient on the treatment delivery day while the patient is in the treatment position. The algorithm consists of two steps. In the first step, PCA motion models are derived by performing PCA on the DVFs resulting from applying DIR on the input 4D-CBCT images. In the second step, an iterative optimization approach is applied on the motion model to generate a sequence of 3D images using CBCT projections. The estimated fluoroscopic 3D images are assessed by localizing the tumor in generated images and comparing these locations to the tumor ground truth location in the CBCT projections. The tumor MAE along the SI direction was 2.29 mm with a 95th percentile of 5.79 mm for patient #1, and 1.89 mm with a 95th percentile of 4.82 mm for patient #2. The clinical applications of this work include image guidance, patient positioning, and delivered dose estimation and/or verification.

## Figures and Tables

**Figure 1 jimaging-08-00017-f001:**
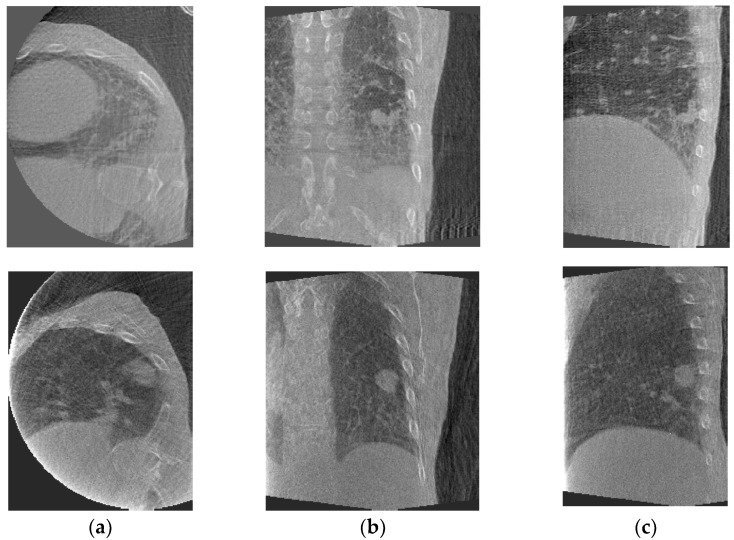
Sample phase (peak-exhale) 4D-CBCT from patient #1 (**top**) and patient #2 (**bottom**): (**a**) axial, (**b**) coronal, and (**c**) sagittal slices.

**Figure 2 jimaging-08-00017-f002:**
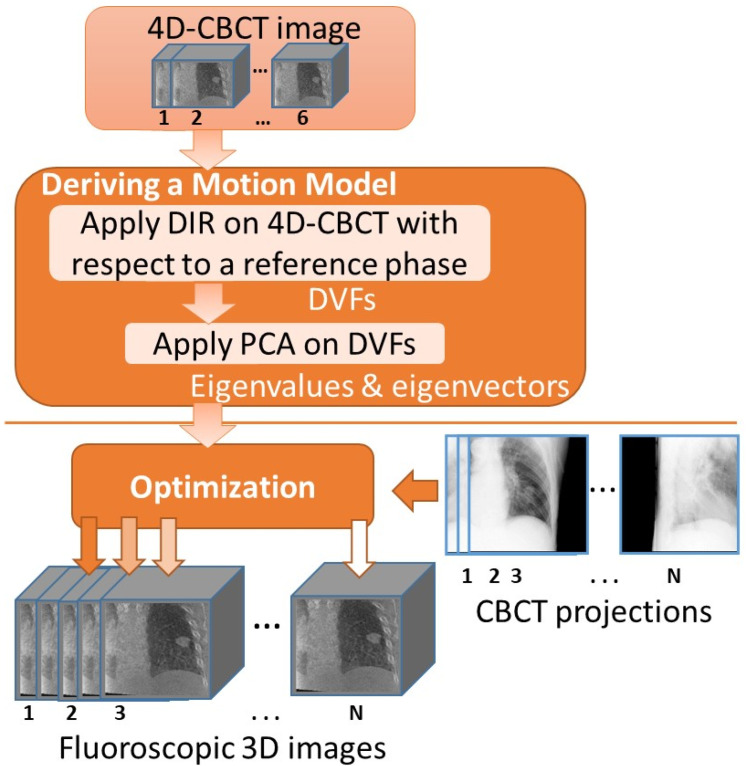
Flowchart of the fluoroscopic 3D image estimation algorithm.

**Figure 3 jimaging-08-00017-f003:**
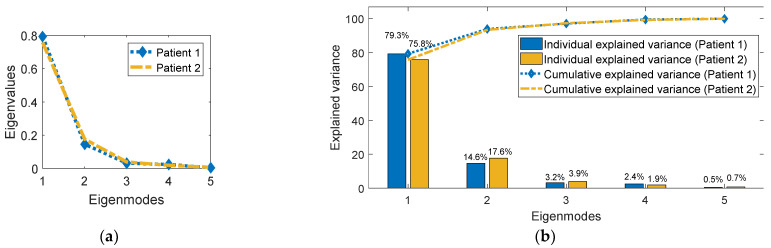
Variance explained by eigenvectors: (**a**) eigenvalues’ spectrum of the motion models of patient #1 and patient #2; (**b**) explained variance ratio of the motion models of patient #1 and patient #2.

**Figure 4 jimaging-08-00017-f004:**
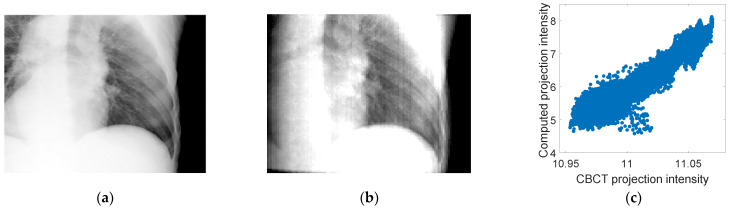
(**a**) Sample CBCT projection from patient #2, (**b**) the corresponding computed projection using the motion model and the 4D-CBCT reference phase, and (**c**) a scatter plot showing the correlation between the intensities of the two images in (**a**,**b**). A linear correlation was found between the two image intensities with a correlation coefficient of 96%.

**Figure 5 jimaging-08-00017-f005:**
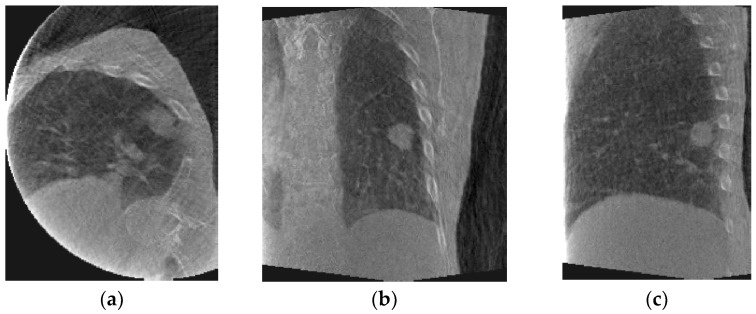
Sample estimated fluoroscopic 3D image from patient #2 dataset: (**a**) axial, (**b**) coronal, and (**c**) sagittal slices.

**Figure 6 jimaging-08-00017-f006:**
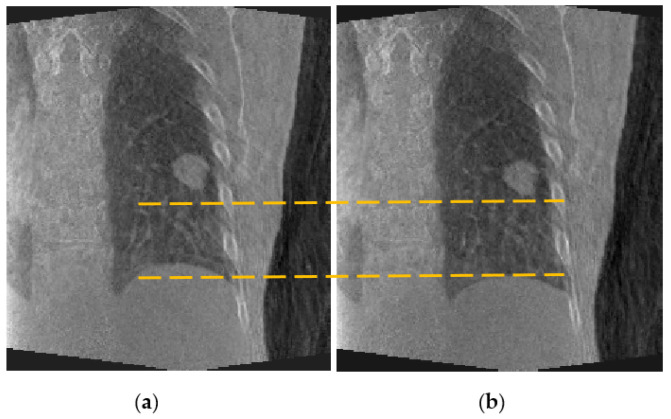
Coronal slices of two estimated fluoroscopic 3D images from patient #2 dataset at different breathing phases: (**a**) exhale phase and (**b**) inhale phase.

**Figure 7 jimaging-08-00017-f007:**
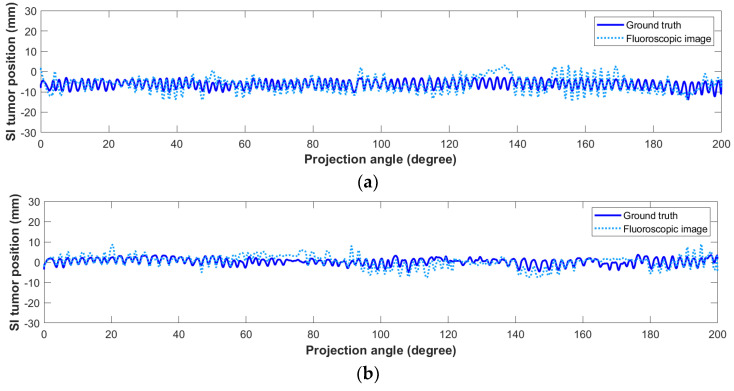
SI tumor position in the estimated fluoroscopic 3D images using motion models derived from 4D-CBCT images of patient #1 (**a**) and patient #2 (**b**).

## Data Availability

The data presented in this study is not publicly available.

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
