# Peer review of "Fluoroscopic 3D Image Generation from Patient-Specific PCA Motion Models Derived from 4D-CBCT Patient Datasets: A Feasibility Study"

_2313-433X, 2022, doi:10.3390/jimaging8020017_

Round 1

Reviewer 1 Report

  1. How the PCA and optimization method differ from previous studies?
  2. Justify why demon registration method is chosen?
  3. It is recommended to elaborate on the process to reduce the noise and the low quality of 4D CBCT images. Selection of ROI ? (page 5, line 176-183)

The write up: English writing and style is good.

Self citation: There are 13 papers (journals/conferences) that include the main author as part of the authors. Possibly reduce the self citation in particular for conference papers.

Ethical approval: It is suggested to include a brief explanation on the ethical approval regarding the two datasets taken from real patients. 

Turnitin report: 89% similarity (84% is from   https://www.preprints.org/manuscript/202111.0519/v1)

Author Response

Dear Respected Reviewer,

We would like to thank you for your valuable comments and suggestions. The manuscript has been revised and benefited substantially from these insightful suggestions, for which we are grateful. We have made every attempt to accommodate your comments, and we have hopefully addressed all of them satisfactorily.

Point-by-point replies to all comments have been added. Please see the attachment below. The manuscript has been revised accordingly with changes tracked as requested.

Reviewer 2 Report

You made a great work in terms of methodology and the paper sounds scientific and well written.

However, some improvements are mandatory before acceptance.

Author Response

Dear Respected Reviewer,

We would like to thank you for your valuable comments and suggestions. The manuscript has been revised and benefited substantially from these insightful suggestions, for which we are grateful. We have made every attempt to accommodate your comments, and we have hopefully addressed all of them satisfactorily.

Point-by-point replies to all comments have been provided. Please see the attachment below. The manuscript has been revised accordingly with changes tracked as requested.
